# Silencing of Odorant-Binding Protein Gene *OBP3* Using RNA Interference Reduced Virus Transmission of Tomato Chlorosis Virus

**DOI:** 10.3390/ijms20204969

**Published:** 2019-10-09

**Authors:** Xiao-Bin Shi, Xue-Zhong Wang, De-Yong Zhang, Zhan-Hong Zhang, Zhuo Zhang, Ju’E Cheng, Li-Min Zheng, Xu-Guo Zhou, Xin-Qiu Tan, Yong Liu

**Affiliations:** 1Hunan Academy of Agricultural Sciences, Institute of Plant Protection, Changsha 410000, China; xiaobin.s@163.com (X.-B.S.); shirley86111@163.com (X.-Z.W.); dyzhang78@163.com (D.-Y.Z.); zhangzhuo@hunaas.cn (Z.Z.); lmzheng66@126.com (L.-M.Z.); tanxinqiu2008@163.com (X.-Q.T.); 2Hunan Academy of Agricultural Sciences, Institute of Vegetable, Changsha 410000, China; 3Department of Entomology, University of Kentucky, Lexington, KY 40546, USA; xuguozhou@uky.edu

**Keywords:** *Bemisia tabaci*, tomato chlorotic virus, volatile organic compounds, odor binding proteins, RNA interference, feeding preference

## Abstract

Tomato chlorosis virus (ToCV) is widespread, seriously impacting tomato production throughout the world. ToCV is semi-persistently transmitted by *Bemisia tabaci* (Gennadius) (Hemiptera: Aleyrodidae). Currently, insect olfaction is being studied to develop novel pest control technologies to effectively control *B. tabaci* and whitefly-borne virus diseases. Despite current research efforts, no report has been published on the role of odorant-binding proteins (OBPs) in insect preference under the influence of plant virus. Our previous research showed that viruliferous *B. tabaci* preferred healthy plants at 48 h after virus acquisition. In this study, we determined the effect of OBPs on the host preference interactions of ToCV and whiteflies. Our results show that with the increase in acquisition time, the OBP gene expressions changed differently, and the *OBP3* gene expression showed a trend of first rising and then falling, and reached the maximum at 48 h. These results indicate that *OBP3* may participate in the host preference of viruliferous whiteflies to healthy plants. When the expression of the *OBP3* gene was knocked down by an RNA interference (RNAi) technique, viruliferous Mediterranean (MED) showed no preference and the ToCV transmission rate was reduced by 83.3%. We conclude that *OBP3* is involved in the detection of plant volatiles by viruliferous MED. Our results provide a theoretical basis and technical support for clarifying the transmission mechanism of ToCV by *B. tabaci* and could provide new avenues for controlling this plant virus and its vectors.

## 1. Introduction

Tomato chlorotic virus (ToCV) belongs to the genus *Crinivirus*, family Closteroviridae. ToCV was first discovered in Florida, the United States, in 1996 [1,2], and has seriously impacted tomato production around the world [3,4]. In Asia, since its first discovery in Taiwan in 2004, ToCV has been identified in Beijing, Shandong, Hebei, Henan, Hunan, and other provinces in China [5], accompanied by the outbreak of *Bemisia tabaci* (Gennadius) (Hemiptera: Aleyrodidae). The yield loss of infected tomato could reach up to 75% or the fruit can even be aborted at an early stage [6]. The spread of ToCV depends on *B. tabaci*, such as *Mediterranean* (MED, Q biotype) and *Middle East–Asia Minor 1* (MEAM1, B biotype) [3,7]. Today, MED has become the dominant population in China [8,9]. 

Virus–host–vector interactions have been paid increasing attention. The virus not only directly changes the reproduction and growth of vectors [2,10], but also indirectly changes the behavior of insect vectors through host plants [2,11,12]. Shi et al. showed that healthy *B. tabaci* had a feeding preference for ToCV-infected plants [2], and ToCV infection can cause volatile changes in tomato plants [13]. 

When plants are infected by pathogens or insects, they produce volatile organic compounds (VOCs), such as terpenoids, isoprenoids, acids, and other compounds [14,15], as an induced defensive response. These induced volatiles can affect insect feeding preference [2,16], in which odorant-binding proteins (OBPs) play an important role [17,18]. OBPs are an important class of biologically active molecules for identifying plant VOCs in the olfactory recognition of insects [19,20,21]. To date, many researchers have studied the characteristics of insect OBPs. Previous research demonstrated that the main role of OBPs is to create the attractive odor of plants, which was related to the host location of insects [22], and Guo et al. concluded that OBPs play important roles in the regulation of phase-related behavior in locusts [23]. However, at present, no report has been published about how OBPs affect the selection of insect vectors in participation with plant viruses.

In this study, we analyzed the relative OBP gene expression of *B. tabaci* after feeding on healthy and ToCV-infected tomato plants using real-time quantitative PCR (RT-qPCR), then analyzed the function of OBPs combined with RNA interference (RNAi) technology and verified the function of the genes through whitefly preference and virus transmission experiments. These results provide a theoretical basis and technical support for clarifying the mechanism of ToCV transmission by *B. tabaci* and for designing new ideas for controlling plant virus and its vectors. Our results also further elucidate the function of OBPs in insect olfactory sensation.

## 2. Results

### 2.1. Detection of ToCV

ToCV was detected using RT-PCR after 30 days of inoculation (Figure 1). After sequencing, the similarity between the amplified sequence and the Beijing tomato ToCV isolate (KC887999.1) was 99.0%.

### 2.2. Analysis of OBPs Expression in B. tabaci MED

The qRT-PCR results show that the relative expression of OBP genes changed differently after feeding on ToCV-infected tomato plants at different acquisition access period (AAP; 24, 48, 72, and 96 h; Figure 2). Among the OBP genes, only the expression levels of *OBP3* and *OBP7* showed the trend of first rising and then falling and differed significantly at different times. In *OBP3* gene expression was highest at 48 h AAP, but in *OBP7* gene expression was highest at 24 h APP. 

### 2.3. dsRNA Synthesis

Expectedly, the amplicon size was of 465 bp for *OBP3* and 598 bp for *GFP* (Figure 3, Figure 4).

### 2.4. Expression of OBPs in B. tabaci MED in Response to dsRNA Treatment

The RNA interference of *B. tabaci* MED showed that the expression of *OBP3* decreased to 27.1% after feeding dsOBP3 for two days (Figure 5). The expression level of other relative OBP genes showed that the relative expression levels of *OBP1* and *OBP2* were significantly up-regulated (*OBP1*: *p* = 0.001; *OBP2*: *p* < 0.001), and *OBP5, OBP6*, and *OBP7* were significantly down-regulated (*OBP5*: *p* = 0.007; *OBP6*: *p* = 0.001; *OBP7*: *p* = 0.003). The expression level of *OBP4* showed no difference compared with dsGFP (*OBP4*: *p* = 0.216) (Figure 6).

### 2.5. Preference of MED on ToCV-infected vs Healthy Tomato Plants

Prior to RNAi treatment, the healthy MED (control NVQ) preferred ToCV-infected tomato plants; in the *OBP3* gene knocked down treatment, the whiteflies preferred healthy plants (Figure 7A, *p* < 0.01). After feeding on ToCV-infected plants for 48 h, the viruliferous MED whiteflies prior to RNAi treatments preferred the healthy plants, whereas the viruliferous MED with the *OBP3* gene knocked down showed no preference (Figure 7B, *p* = 0.77). 

### 2.6. ToCV Transmission Rate

The ToCV infection rate of plants transmitted by whiteflies treated with dsGFP was 64.3%, whereas the ToCV infection rate of plants transmitted by whiteflies treated with *OBP3* dsRNA was only 10.5%. The transmission rate was reduced by 83.3% (Figure 8).

## 3. Discussion

Previous results have shown that *B. tabaci* reached the maximum amount of ToCV accumulation in the 48 h AAP during the process of feeding on ToCV-infected plants, and viruliferous whiteflies preferred healthy plants at this time point [2]. In this study, we used RT-qPCR to detect the relative expression of OBPs in whiteflies after feeding on ToCV-infected tomato plants at different AAPs, and the relative expression of *OBP3* at 48 h AAP was found to be the highest. After *OBP3* was knocked down by RNAi, the viruliferous MED showed no preference, and the ToCV transmission rate was reduced by 83.3%. Therefore, we speculate that *OBP3* may participate in changing the preference of MED whiteflies, thereby affecting the ToCV transmission.

The results of RNAi indicate that the target gene interference was highly efficient, which indicates the effectiveness of studying insect OBPs using RNAi. When we silenced the *OBP3* gene of MED by RNAi, viruliferous MED showed no preference and nonviruliferous MED preferred healthy plants rather than ToCV-infected plants. The silencing of *OBP3* can contribute to ToCV control as it reduces the rate of virus transmission by viruliferous whiteflies. When *OBP3* was knocked down, other OBP genes in MED also changed. This phenomenon is called the olfactory compensation effect [24], possibly due to the knockdown of *OBP3* gene expression; thus, the olfactory system of MED regulates the olfactory compensation of other OBP genes. We conclude that *OBP3* plays a decisive role in the selection of MED for susceptible plants, and *OBP1* and *OBP2* had synergistic effects with *OBP3* in identifying host volatiles, whereas *OBP5, OBP6*, and *OBP7* had antagonistic effects with *OBP3*. In our results, *OBP7* expression also peaked at 24 h and also was significantly higher at 48 h compared to 0 h. It would be interesting to study the effect of silencing *OBP7*, and in the future the role of *OBP7* would be detected. Besides, another approach to unequivocally demonstrate the role of *OBP3* would be to over-express it and check its ability to find healthy plants.

Due to their immobility, plants can release VOCs to communicate with each other and with insects, such as predators of herbivores [25,26], and these chemicals also kairomonally attract insects to feed [26,27]. Insects use their highly developed olfactory system to locate hosts using a variety of chemical stimuli in the environment [28,29]. During olfactory signal transduction, OBPs can selectively transport specific odorants to olfactory receptors to trigger insect behavioral responses [30]. Our results demonstrate that the abilities of OBPs to bind to host volatiles are significantly different, and, in general, OBPs synergistically interact in the process of host plant VOC recognition.

The composition and content of plant VOCs may change when infected with plant viruses and insect vectors. Some researchers analyzed the volatile components of southern rice black-streaked dwarf virus (SRBSDV)-infected plants and healthy plants using gas chromatography–mass spectrometry (GC–MS), and 36 kinds of VOCs were detected from the healthy plants and 37 kinds of VOCs were detected from the SRBSDV-infected plants [31]. Compared with non-viruliferous whiteflies, viruliferous whiteflies induced the volatiles thujene and neophytadiene [32]. In plant–virus–insect vector interactions, nonviruliferous insects tend to select virus-infected plants, whereas viruliferous insects prefer healthy plants [33]. Cucumber mosaic virus (CMV) induced an increase in plant volatile emissions, and induced *Myzus persicae* to preferentially feed on the infected plants [34]. Barley yellow dwarf virus (BYDV) can also induce the release of more volatiles from plants and induce prolonged aphid feeding [35]. The behavior of the insect vectors caused virus transmission. Previous results showed that volatile odorant molecules were transported by OBPs [36]. The role of OBPs in plant volatile recognition and pest control by RNAi has been well studied, such as *OBP1*, *OBP2,* and *OBP4* [37,38,39]. In our research, we determined the role of *OBP3* in whitefly host preference and confirmed that the preference of viruliferous MED for healthy plants is related to OBPs such as *OBP3*, which can be modified by ToCV and can respond to one or more attractive plant volatiles released by healthy plants. 

We studied the mechanism of OBPs affecting the selection of infected plants of MED, providing a new theoretical basis for monitoring and behavioral regulation of MED. This is only the early stage of studies on the transmission of ToCV by MED, and further research is needed to investigate the interaction between OBPs and plant volatiles. In the future, an in-depth study could be conducted on the function and host location of OBPs in *B. tabaci* from the following two aspects: the interaction between related OBPs in *B. tabaci* and developing new attractants for trapping to control *B. tabaci*. Eight proteins are currently known to be involved in the olfactory conduction process of *B. tabaci* [37]. According to current research, a clear division of labor exists between different proteins with a synergistic interaction between them. At present, the relationship and interaction between different proteins are still unclear. Future research is needed to better understand the synergistic interaction between different proteins by implementing technologies such as RNAi. In this study, OBPs affecting *B. tabaci* were selected using RNAi technology. In the next step, the differences in VOCs between healthy plants and ToCV-infected plants can be screened using GC–MS, and field trapping experiments using candidate attractants can be conducted to validate new lures that could potentially be used to control whitefly and ToCV. 

## 4. Materials and Methods

### 4.1. Plants and ToCV

Tomato plants, *Solanum lycopersicum* Mill cv. Zhongza No. 9 (Chinese Vegetable Industry Technology Co. Ltd., Beijing, China), were grown in a potting mix (a mixture of vermiculite, peat moss, organic fertilizer, and perlite in a 10:10:10:1 ratio by volume) in insect-free cages (60 × 60 × 60 cm) in a greenhouse. ToCV-infected plants were produced by agro-inoculation at the 2–4 true leaf stage with *Agrobacterium tumefaciens*-mediated ToCV clones obtained from Zhou Tao, Plant Protection Department of China Agricultural University (Beijing, China). To obtain virus-free plants for the control treatments, mock-inoculated tomato plants receiving the same treatment were used. To confirm inoculation, plant viruses were detected by RT-PCR using gene-specific primers of ToCV heat shock protein 70 (HSP70) sequence (Table 1) at 30 days post-inoculation. 

### 4.2. B. tabaci MED Population

The MED population was cultured on healthy tomato plants in climate-controlled cubicles at 26 ± 2 °C, 60 ± 10% relative humidity (RH), and a photoperiod of 14:10 (light to dark). MED was identified by its biotype using *mitochondrial cytochrome oxidase I* (*mtCOI*) genes (Table 1) about every two months. To obtain the viruliferous MED population, about 2000 healthy MED whiteflies, starved for 4 h, were placed in clip cages on ToCV-infected tomato plants, with 50 whiteflies per clip cage. After an acquisition access period (AAP) of 24, 48, 72, and 96 h, 50 adults were collected randomly from the clip-cages on ToCV-infected leaves and there were 4 replicates at each AAP. The collected samples were stored at −80 °C for gene expression analysis. 

### 4.3. Analysis of OBPs Expression in B. tabaci MED

OBP gene expression was determined at 24, 48, 72, and 96 h post ToCV-inoculation. Total RNA was extracted from 50 adults using the Trizol Invitrogen (Solarbio Science and Technology, Beijing, China), and 1.0 μg of RNA was used to synthesize the first-strand cDNA using a cDNA synthesis kit (TransGen Biotech, Beijing, China). The 25.0 μL reaction system contained 10.0 μL RNase-free H_2_O, 1.0 μL cDNA, 12.5 μL of SYBR qPCR master mix (Vazyme, Nanjing, China), and 0.75 μL of each primer (Table 2) [40]. The relative quantities of mRNA, normalized to reference genes *EF-1α* and *Actin*, were calculated using the comparative cycle threshold (Ct) (2^−ΔΔCt^) method. Three biological replicates and four technical replicates were analyzed. The amplification procedure was as follows: holding stage at 50 °C; 95 °C for 5 min; 40 cycles of 95 °C for 15 s; and 60 °C for 34 s; and a continuous melting curve stage of 95 °C for 15 s, 60 °C for 1 min, 95 °C for 30 s, and 60 °C for 15 s.

### 4.4. dsRNA Synthesis

The *OBP3* gene was amplified using the cDNA of healthy MED as a template using a specific primer containing the T7 promoter sequence (Table 3). The *GFP* gene sequence was amplified by laboratory-preserved GFP PCR products and primers (Table 3) and the PCR reaction system contained 37.9 μL ddH_2_O, 0.5 μL cDNA, 5.0 μL 10× Trans Start Top Taq Buffer (TransGen Biotech, Beijing, China), 4.0 μL dNTPs, 1.0 μL TransStart Top Taq DNA Polymerase (TransGen Biotech, Beijing, China), and 0.8 μL of each primer (Table 3). The PCR amplification procedure was as follows: 95 °C for 5 min; 35 cycles of 95 °C for 30 s, 55 °C for 30 s, 72 °C for 1 min; and 72 °C for 10 min. Next, 5 μL of PCR product was detected by 1.0% agarose gel electrophoresis. Then, the PCR products were purified using an EasyPure PCR product purification kit (Hueyueyang Biotechnology, Beijing, China). The products were quantified using a NanoDrop 2000 (Thermo Fisher Scientific, Beijing, China) and sent to Sangon Biotech (Shanghai, China) for sequencing. The correct PCR products were amplified as a template for synthesizing dsRNA. dsRNA was synthesized and purified using purified PCR products following T7 RiboMAX™ Express RNAi System (Promega, Madison, USA) according to the manufacturer′s instructions, with the correctly sequenced PCR products as templates. Next, 5 μL of dsRNA was detected using 1.0% agarose gel electrophoresis to evaluate the integrity, and the concentration was detected with a NanoDrop 2000 (Thermo Fisher Scientific, Beijing, China). The dsRNA was stored at −80 °C until further use.

### 4.5. Expression of OBPs in B. tabaci MED in Response to dsRNA Treatment

RNA interference was adopted by directly feeding dsRNA to whitefly adults in a feeding chamber [41]. For RNA interference, adult whiteflies were fed 30% sucrose and 5% yeast extract with 500 ng/μL dsOBP3 [41,42]. The artificial diet containing 500 ng/μL dsGFP was used as a negative control. 

Whiteflies were first fed on an artificial diet with dsRNA for 2 days and then transferred to healthy tomato plants. Each treatment was assayed in triplicate. After dsRNA treatment, OBPs expression in MED was measured using RT-qPCR. This method was referred to in Section 4.3.

### 4.6. Preference of MED on ToCV-Infected vs Healthy Tomato Plants

The feeding preference of whiteflies treated with *dsOBP3* for 2 days was determined. ToCV-infected tomato plants and healthy control tomato plants were selected randomly from clean cages, with two plants (one ToCV-infected and one control) per cage. The two plants were placed at a distance of 20 cm from each other. Each experiment was repeated 5 times in 5 cages. After 2 days of dsRNA treatment, about 100 whiteflies were transferred to a clean 1.5 mL centrifuge tube and the tube was placed at the center of the two plants. Then, the lid of the centrifuge tube was opened to release the whiteflies. The number of MED on each plant was investigated after 24 h (ensuring that the MED had landed). The feeding preference of ToCV-infected tomato plants and healthy tomato plants treated with dsGFP for 2 days was used as control.

### 4.7. ToCV Transmission Rate

ToCV was transmitted to plants by whiteflies treated with dsGFP and dsRNA. After an acquisition time of 48 h, the two kinds of whiteflies were collected to clip-cages (with 5 female whiteflies per clip-cage) and attached to healthy plants with 3–4 true leaves. After 48 h of inoculation, the whiteflies were removed from the inoculated plants. After 40 days, the inoculated plant viruses were detected by RT-PCR using the gene-specific primers of ToCV heat shock protein 70 (HSP70) sequence (Table 1). Each treatment was repeated 50 times, for 100 plants in total. Finally, the infection rate was calculated according to the RT-PCR result.

### 4.8. Data Analysis

Data were analyzed using SPSS version 20.0 (SPSS Inc., Chicago, IL, USA). Differences between OBP expression in MED whiteflies after feeding on ToCV-infected tomato plants and healthy tomato plants were compared using two-way analysis of variance (ANOVA). A *t*-test was used to compare the silencing efficiency of RNAi and ToCV transmission rates. The expression of OBPs in MED in response to the dsRNA treatment were compared using two-way ANOVA, and the preference of MED for ToCV-infected vs healthy tomato plants was also analyzed using two-way ANOVA. 

## Figures and Tables

**Figure 1 ijms-20-04969-f001:**
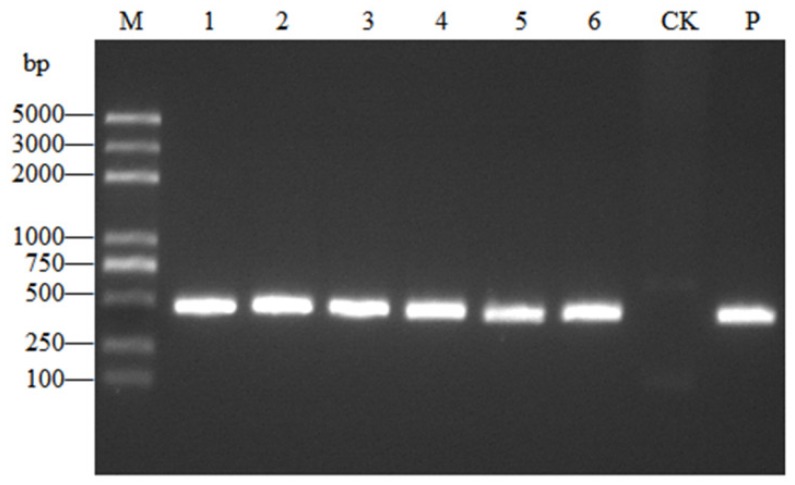
Detection of tomato chlorosis virus (ToCV) in tomato plants by real-time PCR (RT-PCR). M: DNA 2K plus marker; 1–6: Samples 1–6; CK: Negative control; P: Positive control, bp: base pairs.

**Figure 2 ijms-20-04969-f002:**
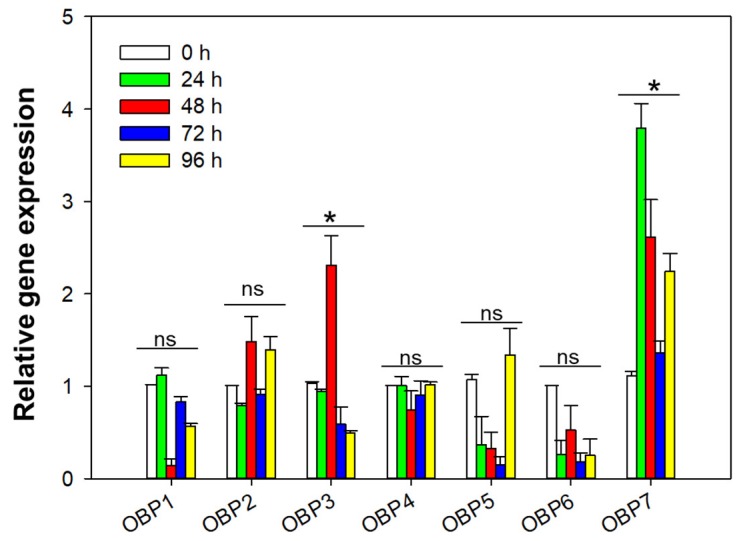
Odorant-binding protein (OBP) expression in *Bemisia tabaci* MED after feeding on ToCV-infected tomato plants at different times. Values represent means ± standard error (SE) for three biological replicates. * *p* < 0.05; *n* = 3.

**Figure 3 ijms-20-04969-f003:**
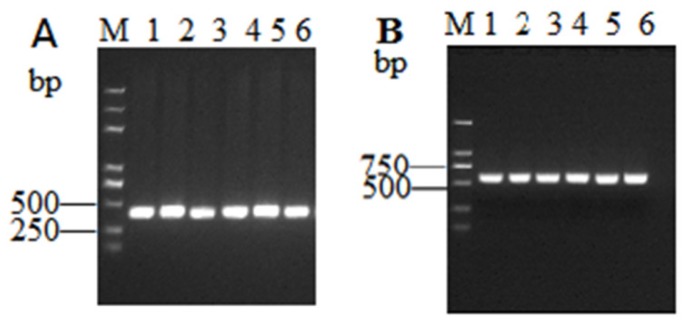
Agarose gel analysis of PCR products of *OBP3* and *Green Fluorescent Protein* (*GFP)* gene fragments. M: DNA 2K marker; 1–6: RT-PCR products of the (**A**) *OBP3* gene and (**B**) *GFP* gene.

**Figure 4 ijms-20-04969-f004:**
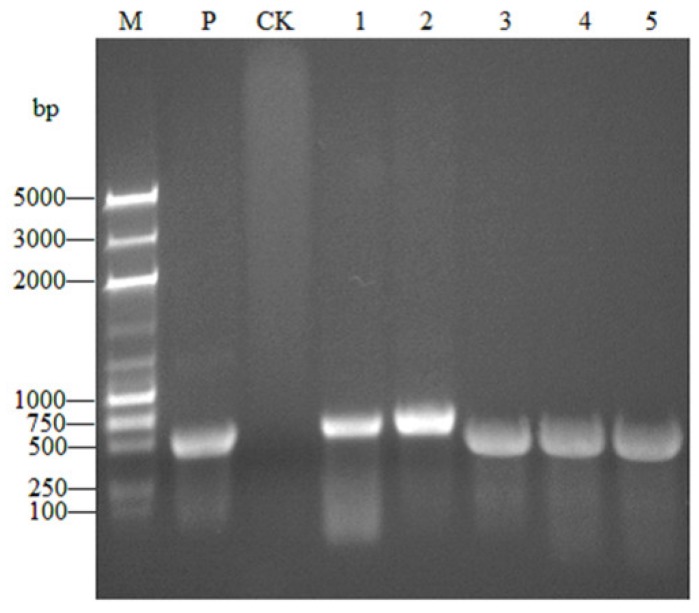
Integrity of double-stranded *OBP3* (*dsOBP3*) and *dsGFP* synthesized in vitro by agarose gel analysis. M: DNA 2K plus marker; P: Positive control; CK: Negative control; 1–2: dsGFP; 3–5: dsOBP3.

**Figure 5 ijms-20-04969-f005:**
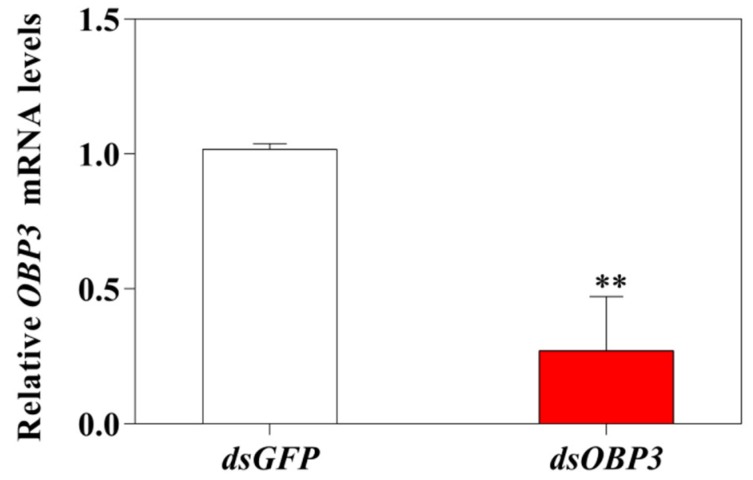
Expression of *OBP3* in *B. tabaci* MED in response to dsRNA treatment. Values are means ± SE. ** *p* < 0.01; *n* = 3.

**Figure 6 ijms-20-04969-f006:**
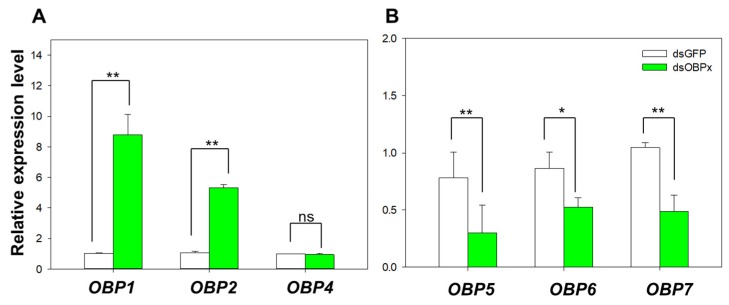
Expression of other OBPs in MED in response to dsRNA treatment. (**A**) Expression of *OBP1*, *OBP2*, and *OBP4* in MED; (**B**) Expression of *OBP5*, *OBP6*, and *OBP7* in MED. Values are means ± SE. ** *p* < 0.01; * *p* < 0.05; ns indicates no significant difference; *n* = 3.

**Figure 7 ijms-20-04969-f007:**
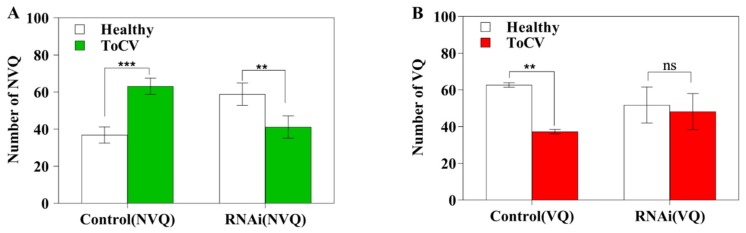
Preference of MED on ToCV-infected vs healthy tomato plants. (**A**) Feeding preference of non-viruliferous Q (NVQ); (**B**) Feeding preference of viruliferous Q (VQ). Control (NVQ): healthy NVQ before RNA interference; RNAi (NVQ): healthy NVQ with *OBP3* gene knocked down; Control (VQ): viruliferous VQ before RNAi; RNAi (VQ): viruliferous VQ with *OBP3* gene knocked down. Healthy: Healthy tomato plants without ToCV infection. ToCV: Tomato plants infected with ToCV. *** *p* < 0.001, ** *p* < 0.01; ns indicates no significant difference; *n* = 5.

**Figure 8 ijms-20-04969-f008:**
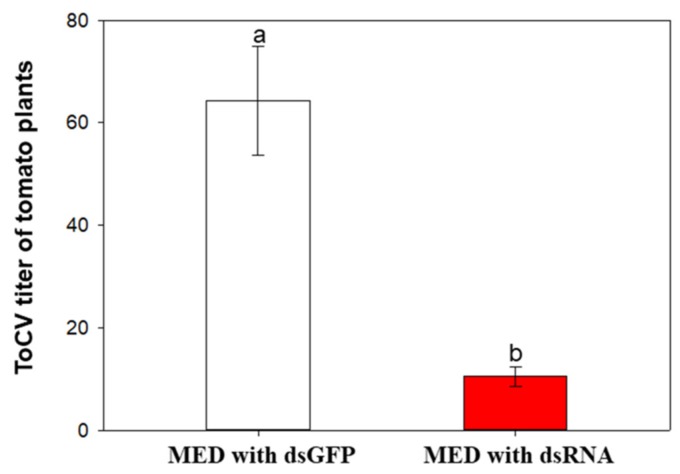
ToCV transmission rate. MED with dsGFP: tomato plants transmitted by whiteflies treated with dsGFP; MED with dsRNA: tomato plants transmitted by whiteflies treated with dsRNA of *OBP3* gene knocked down. Different lowercase letters indicate significant differences at *p* < 0.05, *n* = 50.

**Table 1 ijms-20-04969-t001:** Information of tomato chlorosis virus (ToCV) RT-qPCR and *B. tabaci* biotype primers.

Primer Name	Primer Sequence (5′–3′)	Purpose
ToCV-6	AAACTGCCTGCATGAAAAGTCTC	ToCV RT-PCR
ToCV-5	GGTTTGGATTTTGGTACTACATTCAGT	
tocvhsp70-1	TGTCGAAAGTACCGCCACC	ToCV RT-PCR
tocvhsp70-2	GCTTCCGAAACTCCGTCTTG	
Cl-J-2195	TTGATTTTTTGGTCATCCAGAAGT	*Bemisia tabaci* biotype
R-BQ-2819	CTGAATATCGRCGAGGCATTCC	

**Table 2 ijms-20-04969-t002:** Information of RT-qPCR primers.

Primer Name	Primer Sequence (5′–3′)
BtabOBP1-F BtabOBP1-R	AAGTGCTTGACGGATTATTAC GCATCATATTATCGCAGTGT
BtabOBP2-F BtabOBP2-R	CTCTTATTGGTCTATTTCTCGTT CTTCTTCTTCTGGCATTGG
BtabOBP3-F BtabOBP3-R	CTATCTCGGTTCAGTTCCA TGTCTTTCCACTCGCTAT
BtabOBP4-F BtabOBP4-R	GTTTCTTGGAGTGCGTTTA TCATCATCATCAGCCTCTT
BtabOBP5-F BtabOBP5-R	AAGTAAAGGCTGTGGATGA CGAGTAATAGTTGTTGTCTTGA
BtabOBP6-F BtabOBP6-R	GTAGCAATACAGGTGGAGA ATGACACTCTTGACATTAGC
BtabOBP7-F BtabOBP7-R	TCGAATCAGATGCAGAGGGTG TATCCGGGGGACTCATTCCA
BtabOBP8-F BtabOBP8-R	TGATGGCGTGTCTTATGA CTGAGGTTGAGTGCTGTA
BtabActin-F BtabActin-R	TCTTCCAGCCATCCTTCTTG CGGTGATTTCCTTCTGCATT
BtabEF-1α-F BtabEF-1α-R	TAGCCTTGTGCCAATTTCCG CCTTCAGCATTACCGTCC

**Table 3 ijms-20-04969-t003:** Information of dsRNA synthesis primers.

Primer Name	Sequence (5′–3′)
dsOBP3-R	ATTCTCTAGAAGCTTAATACGACTCACTATAGGGATTGAACCAGCCAAGCTCCC
dsOBP3-F	ATTCTCTAGAAGCTTAATACGACTCACTATAGGGATGATGGATCTCAAAGCTATTTTGCTC
F-086	TAATACGACTCACTATAGGGTTCAGTGGAGAGGGTGAAGGT
R-612	TAATACGACTCACTATAGGGTGTGTGGACAGGTAATGGTTG

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
