# Peer review of "Silencing of Odorant-Binding Protein Gene OBP3 Using RNA Interference Reduced Virus Transmission of Tomato Chlorosis Virus"

_ijms, 2019, doi:10.3390/ijms20204969_

Round 1

Reviewer 1 Report

The manuscript presents an interesting set of results that could explain the behavior of whitefly feeding on virus infected tomato plants.  The authors have done a nice job of examining the potential affect of Odorant Binding Proteins (OBP) on controlling the response to plant volatiles released by the virus infection.  One limitation with the data is that RNAi knockdown of the target OBP3 changes the expression of several additional OBPs making the conclusions less clear.  However the authors are aware of this issue and address it in the discussion.

Author Response

Reviewer 1

The manuscript presents an interesting set of results that could explain the behavior of whitefly feeding on virus infected tomato plants. The authors have done a nice job of examining the potential affect of Odorant Binding Proteins (OBP) on controlling the response to plant volatiles released by the virus infection. One limitation with the data is that RNAi knockdown of the target OBP3 changes the expression of several additional OBPs making the conclusions less clear. However the authors are aware of this issue and address it in the discussion.

Response: Thanks very much for the reviewer’s good comments.

We appreciate your efforts in obtaining review of this manuscript in a timely fashion. I believe that the revised manuscript incorporates most of the suggested changes and conforms to journal guidelines. Please contact me if further modifications are required.

Reviewer 2 Report

Review of manuscript ijms-584249 “Silencing of odorant binding protein gene OBP3…Tomato chlorosis virus”

General Comments:

This is an interesting manuscript on the role of odorant binding protein in the whitefly Bemisia tabaci infected with Tomato chlorosis virus in selecting healthy plants. While interesting there are several aspects of the work that either need to be clarified or additional experiments need to be conducted to fully elucidate the mechanistic role of OBP3 in virus transmission. Also, the language would benefit from being edited by a professional English language service provider. Some of my concerns are as follows:

Specific Comments:

There are some abbreviations in the text that need to be spelled out or clarified upon first usage: MED, MEAM1, AAP for instance. What was the viral titer of ToCV in Bemisia tabaci which elicited the maximal expression of OBP3? From Figure 1, I see that the authors determined ToCV after 20-30 days of inoculation, can ToCV be detected 48h after inoculation? Also, one needs to be precise in determining the actual time of detection of virus in infected plants, 20-30 days is pretty broad range. Why was ToCV not estimated in tabaci? From Figure 2, I see that OBP7 expression also peaked at 24h and also was significantly higher at 48h compared to 0h. It would be interesting to see the effect of silencing OBP7. Another approach to unequivocally demonstrate the role of OBP3 would be to over-express it and check its ability to find healthy plants. I am curious why it chooses healthy plants if the volatiles are emitted by the infected ones (Line 109-113)? There needs to be some explanation for this. If OBP3 gene knock-down resulted in whiteflies preferring healthy plants, it is counter-intuitive since this is helping spread the virus to healthy plants, right?

Author Response

Reviewer 2

General Comments:

This is an interesting manuscript on the role of odorant binding protein in the whitefly Bemisia tabaci infected with Tomato chlorosis virus in selecting healthy plants. While interesting there are several aspects of the work that either need to be clarified or additional experiments need to be conducted to fully elucidate the mechanistic role of OBP3 in virus transmission. Also, the language would benefit from being edited by a professional English language service provider. Some of my concerns are as follows:

Response: Thanks very much for the reviewer’s good comments. The English editing service of MDPI was used for checking grammar, spelling, punctuation and some improvement of style.

Specific Comments:

There are some abbreviations in the text that need to be spelled out or clarified upon first usage: MED, MEAM1, AAP for instance.

Response: The abbreviations in the text were spelled out.

What was the viral titer of ToCV in Bemisia tabaci which elicited the maximal expression of OBP3?

Response: At 48 h AAP, the gene expression of OBP3 in MED reached the maximum, and at this time, the viral titer in viruliferous B. tabaci was 65 times higher than that in non-viruliferous B. tabaci.

From Figure 1, I see that the authors determined ToCV after 20-30 days of inoculation, can ToCV be detected 48h after inoculation? Also, one needs to be precise in determining the actual time of detection of virus in infected plants, 20-30 days is pretty broad range.

Response: ToCV cannot be detected after 48-h inoculation, and after 30-days inoculation ToCV was detected.

Why was ToCV not estimated in tabaci? From Figure 2, I see that OBP7 expression also peaked at 24h and also was significantly higher at 48h compared to 0h. It would be interesting to see the effect of silencing OBP7.

Response: In our previous article, ToCV was estimated in B. tabaci, and the reference has been added in the manuscript. We agree with the reviewer that it would be interesting to see the effect of silencing OBP7, and this has been added in the discussion.

Another approach to unequivocally demonstrate the role of OBP3 would be to over-express it and check its ability to find healthy plants.

Response: Thanks for the reviewer’s suggestion, and this has been added in the discussion.

I am curious why it chooses healthy plants if the volatiles are emitted by the infected ones (Line 109-113)? There needs to be some explanation for this. If OBP3 gene knock-down resulted in whiteflies preferring healthy plants, it is counter-intuitive since this is helping spread the virus to healthy plants, right?

Response: The viruliferous insect vectors preferred healthy plants has been discussed in line 250-260. In our results, before RNAi, the non-viruliferous whiteflies preferred virus-infected plants, and the viruliferous whiteflies preferred healthy plants, after RNAi, the non-viruliferous whiteflies preferred healthy plants, and the viruliferous whiteflies showed no preference. The change of whitefly preference reduces the probability of virus transmission.

We appreciate your efforts in obtaining review of this manuscript in a timely fashion. I believe that the revised manuscript incorporates most of the suggested changes and conforms to journal guidelines. Please contact me if further modifications are required.

Reviewer 3 Report

The manuscript by Xiao Bin Shi is about the control of crinivirus transmission. The authors developed an interesting approach based onto the basic interactions of vector whiteflies with the expression of the insect proteins. Of the 7 OB gene types that they studied, OBP3 was used as model. Taking into account the role played by crinivirus to induce the expression of the OBPs insect gene to attract insect vector, the authors via the RNAi technology conceptually assessed its expression. I approve that the authors regarded the ToCV infection as a standard phenomenon leading to the discovery of OBP genes (Fig,2). Through their results, the authors could not be omitted to study OB7 that they considered as similar to OB 4! Because they regarded both as  unchanged.  Again let see about the results shown in Fig.6!  Back to the histograms fitting to OB3 and 7 in Fig.2, they apparently differed from the rest. While the conceptual research is relevant, the data analysis are confused.  Of course to compare the gene interference between GFP and OBP3  is making sense however the involvement of OB7 gene should be clarified because there is a difference between the role of dsRNA and RNAi . dsRNA is the precursor RNA sliced by the dicer enzyme and should give rise to high amount of RNAi. So to convince the reviewer, the authors should defend their results in that  direction within their data studies. I cannot accept the publication of this manuscript as it is.

In addition, the manuscript was not correctly written there were many parts wherever it is hard to understand (e.g: p.2, l.72-75; p.3. l-96-99…)

Author Response

Reviewer 3

The manuscript by Xiao Bin Shi is about the control of crinivirus transmission. The authors developed an interesting approach based onto the basic interactions of vector whiteflies with the expression of the insect proteins. Of the 7 OB gene types that they studied, OBP3 was used as model. Taking into account the role played by crinivirus to induce the expression of the OBPs insect gene to attract insect vector, the authors via the RNAi technology conceptually assessed its expression. I approve that the authors regarded the ToCV infection as a standard phenomenon leading to the discovery of OBP genes (Fig,2).

Response: Thanks very much for the reviewer’s affirmative evaluation.

Through their results, the authors could not be omitted to study OB7 that they considered as similar to OB4! Because they regarded both as unchanged. Again let see about the results shown in Fig.6! Back to the histograms fitting to OB3 and 7 in Fig.2, they apparently differed from the rest. While the conceptual research is relevant, the data analysis are confused.

Response: The OBP7 has been reduced according to the data analysis, and the previous error on data analysis has been revised.

Of course to compare the gene interference between GFP and OBP3 is making sense however the involvement of OB7 gene should be clarified because there is a difference between the role of dsRNA and RNAi. dsRNA is the precursor RNA sliced by the dicer enzyme and should give rise to high amount of RNAi. So to convince the reviewer, the authors should defend their results in that direction within their data studies. I cannot accept the publication of this manuscript as it is.

Response: The previous figure legend was confused, which should be dsGFP and dsOBPx, and this has been revised.

In addition, the manuscript was not correctly written there were many parts wherever it is hard to understand (e.g: p.2, l.72-75; p.3. l-96-99…)

Response: The sentences have been revised, and the whole manuscript has been checked to avoid any mistakes.

We appreciate your efforts in obtaining review of this manuscript in a timely fashion. I believe that the revised manuscript incorporates most of the suggested changes and conforms to journal guidelines. Please contact me if further modifications are required.

Round 2

Reviewer 2 Report

The authors have adequately addressed the concerns of the reviewers and the revised version reads much better.

Author Response

Reviewer 3
Reviewer 3 comments:
Over substantial efforts done by the authors, while OBP3 effect doesn’t rise any contest, enclosed are my comments and inputs to improve the final version:
- I recognized the extent of correction achieved, the authors have taken in account the different key-points commented in the former version. In particular the interpretation of the OBP7 pattern that is readily up-and down-regulated. As reported here they didn’t show more research studies but they commented it in the discussion section.
Response: Thanks very much for the reviewer’s good comments.

- Page 6, l.170-175 should be changed because the figure 4 didn’t represent any result studies, it is simply a control. So it should be remoted and replaced to: “Expectedly, the amplicon size was of 465bp for OBP3 and 598bp for GFP (Fig.3).”
Response: The sentences have been changed according to the reviewer’s suggestion.

- Data studies have served virologists, zoologists on plant-insect interactions. Obviously there is an endless quest to seek what would be the innovative strategies to control these pests.
- I can accept the publication of this manuscript according to these forensic remarks and recommandation.
Response: Thanks very much for the reviewer’s good comments. We are trying our best to find more innovative strategies to control the pests.

We appreciate your efforts in obtaining review of this manuscript in a timely fashion. I believe that the revised manuscript incorporates most of the suggested changes and conforms to journal guidelines. Please contact me if further modifications are required.

Sincerely,

Yong Liu, Ph.D, professor
Hunan Plant Protection Institute,
Hunan Academy of Agricultural Sciences,
Changsha 410125, China
E-mail: [email protected]